# Biocompatibility and antimicrobial efficacy of iodine-supported titania nanotubes on 3D-printed Ti-6Al-4V implants

Pariwat Taweekitikul[1,2,3], Abdul Azeez Abdu Aliyu[4], Nanthawan Jinakul[5], Amaraporn Wongrakpanich[6,7], Chedtha Puncreobutr[4,3], Jirapon Khamwannah[8], Boonrat Lohwongwatana[4,8,9], Saran Tantavisut[3,10]*

1 Biomedical Engineering Program, Faculty of Engineering, Chulalongkorn University, Bangkok, Thailand, 2 Department of Orthopedic, Queen Savang Vadhana Memorial Hospital, Chonburi, Thailand, 3 Biomedical Engineering Research Center, Chulalongkorn University, Bangkok, Thailand, 4 M3D Laboratory, Advanced Materials Analysis Research Unit, Department of Metallurgical Engineering, Faculty of Engineering, Chulalongkorn University, Bangkok, Thailand, 5 Department of Microbiology, Faculty of Pharmacy, Mahidol University, Bangkok, Thailand, 6 Department of Pharmacy, Faculty of Pharmacy, Mahidol University, Bangkok, Thailand, 7 Center for Research on Extemporaneous Compounding and Pharmaceutical Technology Innovation, Faculty of Pharmacy, Mahidol University, Bangkok, Thailand, 8 Department of Metallurgical Engineering, Faculty of Engineering, Chulalongkorn University, Bangkok, Thailand, 9 Department of Materials Science and Engineering, Massachusetts Institute of Technology, Cambridge, Massachusetts, United States of America, 10 Center of Excellence in Hip Fracture, Department of Orthopaedics, Faculty of Medicine, Chulalongkorn University, Bangkok, Thailand

* stantavisut@gmail.com

## Abstract

Implant-associated infections (IAI) are a significant concern within the field of orthopedics. To develop an implant with antimicrobial properties, titania nanotubes (TNTs) supported with iodine were synthesized on 3D-printed Ti-6Al-4V implants using the electrochemical anodization (ECA) technique. This study aims to analyze the release profile of iodine and the antimicrobial efficacy and cytotoxicity of the iodine-supported TNT (I-TNTs) on the 3D-printed Ti-6Al-4V implant. 3D-printed Ti-6Al-4V samples were doped with six different iodine formulations, including four test groups containing TNT and two control groups without TNT. As printed 3D Ti-6Al-4V samples and TNTs samples without iodine were also utilized as control groups. *In vitro* assays were performed to assess the drug elution, cytotoxicity, and antimicrobial efficacy. All tested I-TNTs samples exhibited sustained iodine release over 28 days without an initial burst. Notably, the amount of iodine released from I-TNTs was significantly higher compared to the control group. TNTs with a higher aspect ratio (AR) and the ECA process using higher potassium iodide (KI) concentration were found to have better cumulative iodine release profiles. No cytotoxicity was observed when tested against the mouse calvaria-derived preosteoblast cell line (MC3T3-E1). The antibacterial property of the implant surface became evident within 24 hours, with complete inhibition of *S. aureus* and MRSA in I-TNTs samples. This innovative approach is an intriguing alternative for preventing infections on 3D-printed Ti-6Al-4V implants.

**Data availability statement:** All relevant data are within the manuscript and its Supporting Information files.

**Funding:** This work was financially supported by Ratchadapiseksompotch Research Fund (Grant number GA65/80), Chulalongkorn University, Thailand. The funders had no role in the study design, data collection and analysis, decision to publish, or preparation of the manuscript.

**Competing interests:** The authors have declared that no competing interests exist.

## 1. Introduction

Additive manufacturing/3D printing technology offers several advantages for orthopedic implants, enabling the production of personalized prostheses tailored to the patient's specific anatomical need [1,2]. This customization improves the anatomical and functional results, reduces surgical time, and minimizes complications [3]. However, similar to conventional orthopedic implants, 3D printed implants also exhibit a significant complication, namely implant-associated infection (IAI). Existing literature has documented infection rates of 7–12.9% for 3D printed implants [4–7], even when utilizing contemporary infection prevention measures.

To reduce the incidence of IAI, numerous localized drug delivery systems have been proposed, such as applying local antibiotic powder, beads, or modifying implant surfaces to effectively combat infection [8,9]. Various antimicrobial coatings have been utilized on the 3D-printed implants, including gentamicin [10,11], tetracycline [12], minocycline [13], vancomycin [14], clindamycin [15], and chitosan & gelatin [16].

With the global challenge in the era of increasing antibiotic-resistant pathogens [17], the significance of iodine as an intriguing option for antimicrobial coating is emphasized. Iodine possesses various advantages in this regard. Firstly, it exhibits a broad spectrum of antimicrobial activity, effectively eliminating bacteria, viruses, fungi, and spores while avoiding the induction of bacterial resistance [18]. Second, iodine has a proven safety record as a disinfectant and contrast agent [19]. Furthermore, iodine is biologically safe as it is a trace metal and can be excreted by the kidneys if released from the implant. Notably, Shirai *et al.* [18–20] have published a series of literature highlighting the robust antimicrobial properties and excellent biocompatibility of iodine-coated nanoporous layers on titanium implants in laboratory and clinical settings.

Not only are nanoporous surfaces well-known for their ability as drug carriers in drug release applications, but $TiO_2$ nanotubes (TNTs) also possess this characteristic. The tubular morphology and self-organized nanostructures of TNTs have attracted significant attention in implant manufacturing due to their mechanical stability, cost-effectiveness, and exceptional biocompatibility [21,22]. The utilization of anodic oxidation as a straightforward and versatile technique enables the synthesis of TNTs with controlled structure and morphology, rendering it a promising area of exploration for drug delivery research [23].

Novel approaches to mitigate IAI are centered around preventing early postoperative infections, which occur within four weeks of the operation [24]. The "race for the surface" concept, where tissue cells compete with bacteria for implant surface adhesion, is crucial during this period [25]. Antimicrobial coatings directly influence surface properties, hindering bacterial adhesion and biofilm formation [26]. Therefore, surface modification with antimicrobial properties is paramount during this critical phase.

Considering all of these factors, this investigation aims to examine the release profile of iodine over 28 days and assess the antimicrobial efficacy and cytotoxicity of iodine-supported TNTs on 3D-printed titanium implants focusing on different iodine formulations with varying aspect ratio (AR) and potassium iodide (KI) concentration

used in fabrication process. To the best of our knowledge, there has been no report thus far regarding the utilization of TNT as a drug carrier for delivering iodine on 3D-printed titanium implants. The underlying hypothesis was that this particular implant would demonstrate significant antimicrobial effectiveness without any detectable levels of cytotoxicity.

## 2. Materials and Methods

### 2.1. Preparation of iodine-supported TNT

3D printed specimens of Ti-6Al-4V were manufactured by Meticuly Company in Bangkok, Thailand, using SolidWorks 2020 software to design and model the Ti-6Al-4V plate implant (circular discs with a diameter of 10 mm and a thickness of 2 mm). The implant model is converted to an STL file and fabricated using D50 micro size Ti-6Al-4V powder by SLM (Model: Mlab cusing 100R, Concept Laser GmbH, Germany) technique. The electrochemical anodization (ECA) process entailed the utilization of a 0.5 wt.% solution of Ammonium fluoride ($NH_4F$, Sigma-Aldrich, USA), 1.5 wt.% deionized water (DI), and 98 wt.% Ethylene glycol (EG, Qrec, Auckland, New Zealand) as the electrolyte solution. The polished 3D printed Ti-6Al-4V specimen served as the anode, while the platinum plate, positioned 1 cm away, functioned as the cathode. In our previously published work [27] we verified that the optimal TNT's aspect ratio (AR) was achieved with a constant applied voltage of 60 V at durations of 210 (TNT-A) and 240 (TNT-B) minutes. As a result, TNTs are synthesized using these values in the current study utilizing ECA. The TNT was synthesized utilizing the ECA technique with a constant applied voltage of 60 V and varying durations of 210 and 240 minutes (labeled TNT-A and TNT-B, respectively). Subsequently, we carry out a modified ECA process by altering the electrolyte composition to include two different concentrations of potassium iodide (KI, Sigma-Aldrich, USA) at 0.1 (labeled I-TNT A1, I-TNT B1) and 0.5 g/L (labeled I-TNT A5, I-TNT B5). This concentration range was selected considering the fact that, povidone-iodine solutions typically contain 1% available iodine, while KI contains 2% iodine. To achieve the same iodine content as 1.32 g/L of povidone-iodine, approximately 0.66 g/L of KI is required, ensuring that KI does not exceed 1 g/L to avoid potential cytotoxicity to human cells. Therefore, in this study, the KI concentrations of 0.1 g/L and 0.5 g/L for was selected after series of preliminary studies. As printed 3D Ti-6Al-4V samples and TNT-A samples were also utilized as control groups in the antimicrobial efficacy and cytotoxicity assay. For the control groups in the drug elusion assay, as-printed 3D Ti-6Al-4V specimens (without TNT) were utilized in the modified ECA process with KI concentrations of 0.1 and 0.5 g/L and labeled as control-1 and control-5, respectively. The modified ECA processes are conducted using a constant voltage of 60 V, a constant time of 60 minutes under ambient temperature conditions. Fig 1 shows the schematic illustration of the overall methodology for preparing 3D-printed Ti-6Al-4V samples. Table 1 shows conditions in the ECA to prepare six formulations of iodine-doped samples. The morphology and dimensions of nanotubes on anodized surfaces were assessed at 15 kV using field emission scanning electron microscopy (FESEM, FEI Quanta FEG 250, Thermo Fisher Scientific, Oregon, USA). Image J software version Java 8 (National Institutes of Health, USA) was used to measure the dimensions of nanotubes by measuring three spots on each specimen, five times for each spot.

### 2.2. Drug elusion assay

All six formulations of the iodine-doped samples and control samples were placed in well plates containing 5 mL of phosphate-buffered saline. These samples were then stored in an incubator shaker (Mini shaking incubator, Hangzhou Miu Instruments, China) at a temperature of 37˚C and a speed of 100 rpm. At specific time intervals (1, 12, 24 hours, 7, 14, and 28 days), 100 µL of the samples were extracted and subsequently diluted with 10 mL of 1% $HNO_3$. All the samples were then stored in the refrigerator for further analysis. The supernatant obtained at each time point was then quantified for iodine concentration.

The iodine content in the samples was quantified using the inductively coupled plasma mass spectrometer (ICP-MS) methods (iCAP Qc, Thermo Fisher Scientific, Germany), and then the average was calculated for the three individual

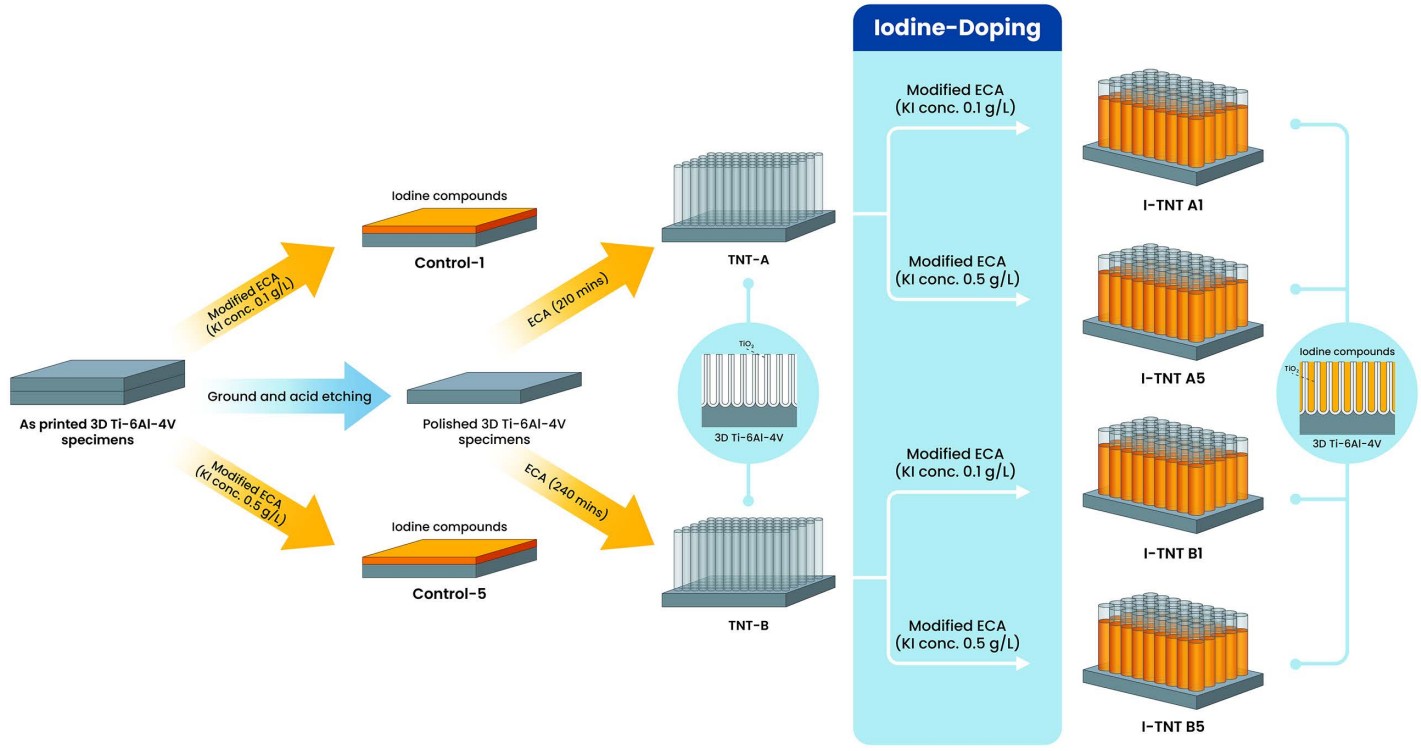

**Fig 1. Schematic illustration of the step-by-step fabrication of iodine-doped 3D printed Ti-6Al-4V specimens in this study. Abbreviations:** ECA, electrochemical anodization; TNT, titania nanotube.

**Table 1. Conditions in the ECA to prepare iodine-doped samples.**

| Formulation Number | Formulation Name | Anode | ECA duration (min) | Electrolyte KI (g/L) | Temperature, Voltage, and Time |
|---|---|---|---|---|---|
| 1 | Control-1 | As-printed 3D Ti | – | 0.1 | Room temperature, 60 V, and 60 minutes |
| 2 | Control-5 | | – | 0.5 | |
| 3 | I-TNT A1 | TNT-A | 210 | 0.1 | |
| 4 | I-TNT A5 | | 210 | 0.5 | |
| 5 | I-TNT B1 | TNT-B | 240 | 0.1 | |
| 6 | I-TNT B5 | | 240 | 0.5 | |

samples in each specific condition. The profiles depicting the cumulative release of iodine from the samples (y-axis) were plotted in relation to time (x-axis). The standard curve for iodine quantification is provided in the S1 Fig.

## 2.3. In vitro cytotoxicity assay

**2.3.1. Tested on liquid extracts of titanium specimens.** The experiments utilized the mouse calvaria-derived preosteoblast cell line (MC3T3-E1, ATCC® CRL-2593™). The cells were cultured in α-minimum essential medium (α-MEM, Gibco®, Thermo Fisher, USA) supplemented with 10% fetal bovine serum (FBS, triple 0.1 μm sterile filtered, Hyclone™, GE Healthcare Bio-Sciences, Austria) and antibiotic-antimycotic (Gibco®, Thermo Fisher, USA). The cells were incubated

   

at a temperature of 37˚C and 5% $CO_2$. Following the recommendations provided by ATCC, the handling and subculturing procedures were conducted with a subcultivation ratio ranging from 1:2–1:4 when the cells reached 80% confluency.

An assessment was performed to determine the cytotoxicity of the specimens in an in vitro setting, following the ISO 10993–5, Biological evaluation of medical devices, Part 5 guideline with some modifications [28]. The specimens used in this study consisted of liquid extracts from four groups of titanium materials, including as printed 3D Ti-6Al-4V, TNT-A, I-TNT A5, and I-TNT B5. Phosphate buffer saline (PBS) solution was employed as the extraction medium. The extraction process involved incubating the titanium specimens in PBS for 24 hours at a temperature of 37˚C within a shaking incubator operating at a speed of 100 rpm.

Cell viability is determined by assessing the metabolic activity of cells, which involves the reduction of the yellow compound MTT (3-(4,5-dimethylthiazol-2-yl)-2,5-diphenyltetrazoliumbromide) to a violet formazan. MC3T3-E1 cells were seeded in 96-well plates at a concentration of $1x10^4$ cells per well and allowed to adhere for 24 hours. Treatments were administered at a 1:1 and 1:3 volume ratio of treatment to medium, with a working volume of 100 µL, for 24 and 48 hours. Following the incubation period, morphological changes were examined under a microscope. The treatments were replaced with the MTT solution (PanReac AppliChem ITW Reagents, Germany) supplement with the serum-free medium at a concentration of 0.5 mg/mL. Following a 2-hour incubation period at 37˚C and 5% CO2, the media were removed. Dimethyl sulfoxide (DMSO, Sigma-Aldrich, USA) was introduced to dissolve the formazan crystals. The absorbance at 570 nm was measured using a microplate reader (ClarioStar Multimode Microplate Reader, BMG Labtech, Germany). Cell viability was calculated as a percentage of the absorbance value obtained from PBS-treated cells, with background correction using a 100 µL DMSO. As a positive control, sodium lauryl sulfate (SLS, S. Tong Chemicals Co., Ltd., Thailand) was utilized [29]. SLS treatments were also performed for 24 and 48 hours.

**2.3.2. Tested on direct contact with titanium specimens.** The sterility of all four groups of titanium specimens was ensured by subjecting each side of the specimen to ultraviolet (UV) radiation for 30 minutes. Each specimen was then placed in a 24-well plate. MC3T3-E1 cells were seeded onto each specimen at a concentration of $2x10^4$ cells per well in 50 µL media. After 15 minutes, an additional 600 µL of media was gently added. Wells containing only MC3T3-E1 cells served as the positive control. Incubation was carried out at 37˚C with 5% $CO_2$ for 7 and 14 days. Media was replaced every 2–3 days. At each time point, the titanium specimens were collected and transferred to a new 24-well plate. The MTT assay was performed, and cell viability was reported as a percentage, following a procedure similar to the test on liquid extracts of titanium specimens.

## 2.4. Determination of antibacterial property

An examination of the antimicrobial properties was conducted in accordance with the guidelines outlined in JIS Z 2801:2010 with some modifications. The experiment involved using four separate groups of titanium specimens, similar to those utilized in the cytotoxicity tests. To ensure sterility, all specimens underwent a surface sterilization process utilizing UV light for 30 minutes on each side. The bacterial strains employed in this particular study consisted of *Staphylococcus aureus* (ATCC 6538) and Methicillin-Resistant *Staphylococcus aureus* (MRSA) (DMST 20654). Before testing, the bacterial samples were pre-incubated in nutrient agar and maintained at 35˚C for 24 hours. Subsequently, a single platinum loop containing the test bacteria was evenly dispersed in a small quantity of 1/500 Nutrient broth (1/500 NB). This inoculum was then diluted until the bacterial concentration reached a range of $2.5x10^5$- $10x10^5$ cfu/mL.

Each titanium specimen was positioned within a sterilized petri dish. Subsequently, 0.1 mL of test inoculum was applied onto each test piece surface. The specimen ($220 mm^2$) was then gently wrapped in a sterile polyethylene film, ensuring even distribution of the inoculum. The inoculated specimens were incubated for 24 hours at 35°C. During the washout process, the covering film and the test piece were placed in a sterilized stomacher pouch. A 10 mL of soybean-casein digest lecithin polysorbate broth (SCDLP) was employed to extract the microbes. The washout process was performed immediately after inoculation and again after a 24-hr incubation period.

Ten-fold serial dilutions of the extract were prepared using PBS. These diluted samples were dispensed onto two sterile petri dishes (1 mL each) and supplemented with 15–20 mL of plate count agar at approximately 45°C. The plates were incubated at 35°C for 24 hours.

The test findings were presented in two manners: firstly, the quantification of viable bacteria (CFU/cm$^2$) (equation 1) with a percentage reduction of viable bacteria at 24 hours (equation 2), and secondly, the determination of antibacterial activity through the utilization of equation 3. This method was repeated three times for each specimen. The antimicrobial efficacy is evaluated based on the standards set by JIS Z 2801. The determination of an antibacterial activity value of 2.0 or higher is taken into account.

$$N = \frac{C x D x V}{A}$$

$$(1)$$

where N: number of viable bacteria (per 1 cm$^2$ of test piece)

 C: count of colonies (average count of colonies of two petri dishes adopted)

 D: dilution factor (that of dilution dispensed into petri dishes adopted)

 V: volume of SCDLP broth used for washout (mL)

 A: surface area of test piece that was exposed to bacteria (cm$^2$)

$$Percentage\ reduction = \frac{A - B}{A}\ x100$$

$$(2)$$

where A: number of viable microorganisms at 0 hr

 B: number of viable microorganisms at 24 hr

$$R = (U_t - U_0) - (A_t - U_0) = U_t - A_t$$

$$(3)$$

where R: antibacterial activity

 $U_0$: average of logarithm number of viable bacteria immediately after inoculation on untreated test pieces

 $U_t$: average of logarithm number of viable bacteria after inoculation on untreated test pieces after 24 hr

 $A_t$: average of logarithm number of viable bacteria after inoculation on antibacterial test pieces after 24 hr

### 2.5. Statistical analysis

Data are expressed as mean ± standard deviation (SD)The statistical significance of the drug elution assay, the quantification of viable bacteria, and the cell viability for the test on direct contact with titanium specimens between groups were determined using One-way ANOVA. The unpaired t-test was performed to compare the quantification of viable bacteria and the cell viability for the test on direct contact with titanium specimens within the same group. All statistical tests were performed using IBM SPSS 25.0 software. A *p*-value less than 0.05 was considered significant.

## 3. Results

### 3.1. Iodine-supported TNT on 3D-printed titanium

This study involved fabricating 3D-printed titanium specimens into six groups: as-printed 3D-printed titanium, titanium nanotube (TNT), I-TNT A1, I-TNT A5, I-TNT B1, and I-TNT B5. A spherical medical-grade commercial Ti-6Al-4V powder was used as the raw material for printing. The TNT fabrication process was done using the ECA process. After that, the modified ECA was performed to integrate iodine on the TNT surface. All specimens were designed as circular discs with a diameter of 10 mm and thickness of 2 mm (Fig 2). Detailed information on the TNT dimensions and surface morphology are shown in Table 2 and Fig 3.

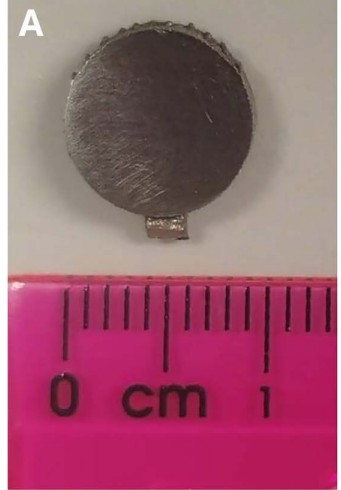 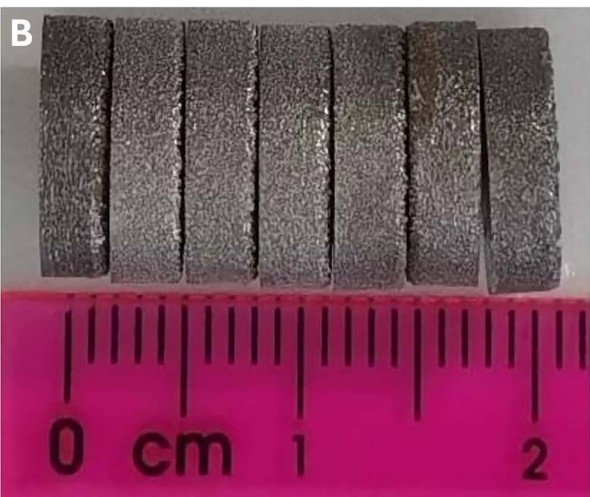

**Fig 2. As printed 3D Ti-6AL-4V specimens. The specimen diameter is equal to 10 mm (A) with a 2 mm thickness (B).**

**Table 2. I-TNT dimensions of each specimen tested in this project.**

| Group | Nanotube Morphology | | |
|---|---|---|---|
| | Pore Diameter (nm) (mean±SD) | Length (nm) (mean±SD) | Aspect Ratio |
| I-TNT A5 | 81.78±33.18 | 6,762.2±317.96 | 82.69 |
| I-TNT B5 | 108.13±22.94 | 6,237.47±1,340.52 | 57.69 |

### 3.2. Iodine release profile from the I-TNTs specimens

The quantitative analysis and characterization of iodine-doped TNTs is reported in our recently published article [27]. Fig 4 illustrates the cumulative release of iodine from the I-TNTs samples and controlled samples. The I-TNTs groups exhibited a continuous release of iodine for the first 7 days (168 hours), followed by a significant decrease in release rate, reaching a plateau phase. In contrast, the control group exhibited a much lower overall release. The burst release phase in the control samples was shorter, ending after 24 hours and entering the plateau phase earlier than the I-TNTs samples. Notably, the cumulative iodine release profiles were significantly higher in the I-TNTs groups compared to the control group (p=0.000).

There was no significant difference in iodine release profiles between the different groups of control samples at each time point. However, for the I-TNTs samples, significant differences were observed as early as 1 hour, with I-TNT A5 exhibiting higher release than I-TNT B1 and I-TNT B5 (p=0.005 and 0.024, respectively). As time increased, the differences between each group became more pronounced. The first significant difference between I-TNT A1 and I-TNT A5 was observed at 24 hours (p=0.014), while I-TNT B1 and I-TNT B5 showed significant differences at the final time-point of 672 hours (p=0.000). I-TNT A5 consistently maintained the highest level of iodine release compared to the other groups from 24 hours until the end of the observation period.

### 3.3. In vitro cell cytotoxicity

The cytotoxicity of I-TNTs and its degradation products was evaluated using an MTT assay. Two experiments were conducted. Firstly, MC3T3-E1 cells were exposed to media obtained from I-TNTs extractions for 24 hours to assess the impact of released substances from the materials. The second experiment involved direct cell contact with I-TNTs

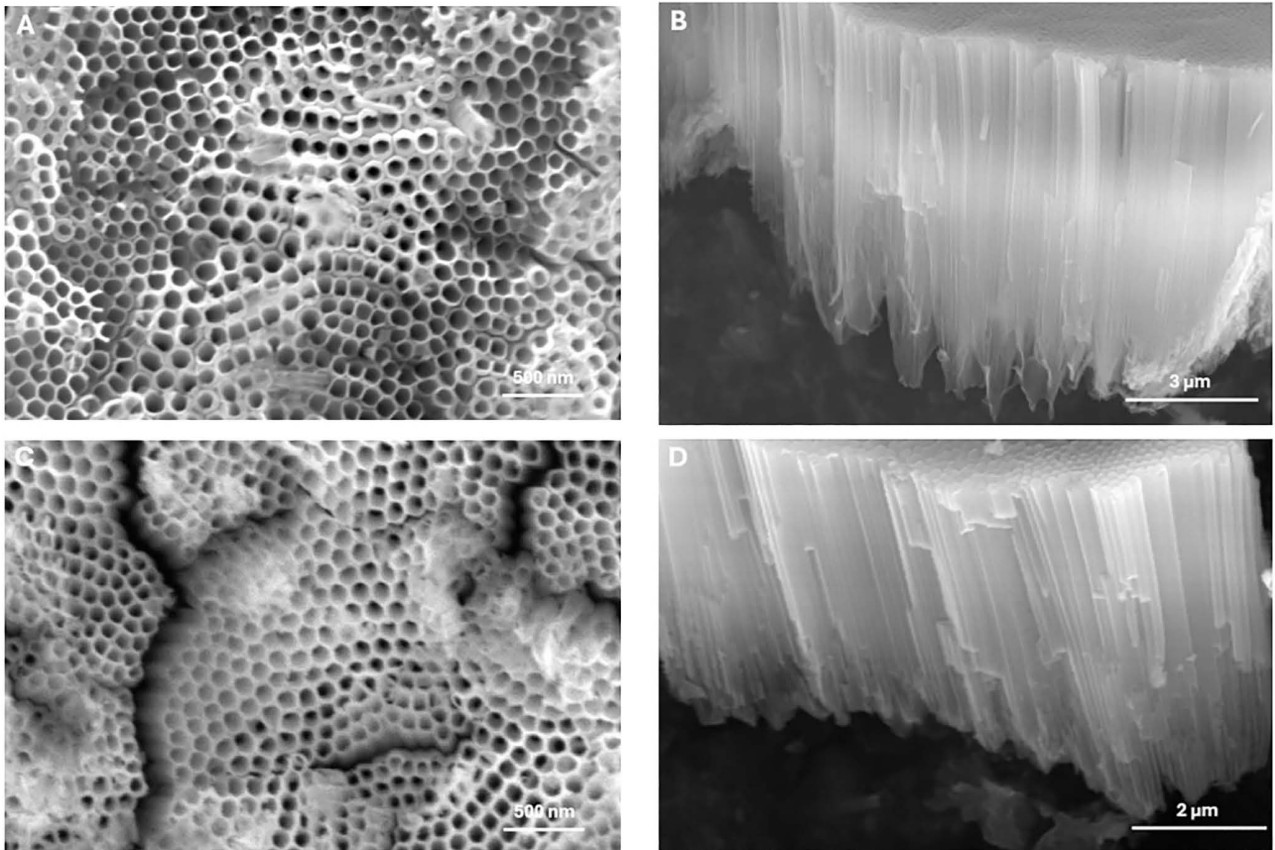

**Fig 3. The FESEM micrographs represent the top (A, C) and side views (B, D) of the I-TNT A5 (A, B) and I-TNT B5 (C, D).**

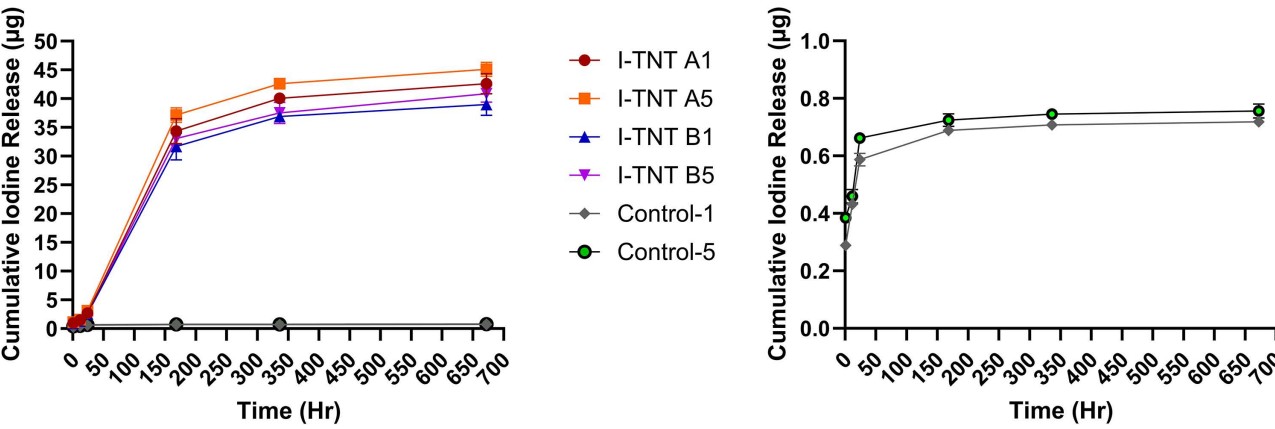

**Fig 4. The cumulative release profiles of iodine from the I-TNTs samples and controlled samples.** The iodine release was monitored for 28 days. Data are expressed as mean ± SD, n = 3. One-way ANOVA was used to determine statistically significant difference.

surfaces to evaluate cytotoxicity from the material and potential surface-released components. For the qualitative morphological assessment of the extracts' cytotoxicity, all samples were assigned a score of 0, indicating no reactivity. There was no evidence of cell lysis or decrease in cell proliferation, and no intracytoplasmic granules were observed. Furthermore, all tested samples exhibited cell viability exceeding 70%, indicating a lack of cytotoxic effect according to ISO 10993−5 (Fig 5). In this particular experiment, a positive control using SLS displayed concentration-dependent cytotoxicity (S2 Fig).

For direct contact with titanium samples, as printed 3D Ti-6Al-4V exhibits a higher proportion of viable cells compared to TNT-A, I-TNT A5, and I-TNT B5 at both 7 days and 14 days. However, there was no statistically significant difference among TNT-A, I-TNT A5, and I-TNT B5 at either time point. Within each group, comparing cell viability at 7 and 14 days reveals that as printed 3D Ti-6Al-4V, I-TNT A5, and I-TNT B5 demonstrate a greater percentage of viable cells at 14 days, with no significant difference observed (39.03% VS 47.29% (p = 0.502), 13.13% VS 14.12% (p = 0.626), and 11.67% VS 11.7% (p = 0.99)). TNT-A samples showed a slight decrease in viable cells at 14 days (11.61% VS 9.13%), but the difference was not statistically significant (p = 0.263) (Fig 6).

### 3.4. Antibacterial property

*S. aureus* [30,31] and MRSA [32] are the most prevalent pathogens in orthopaedic surgical site infections. The antimicrobial activity of I-TNT specimens was assessed by subjecting *S. aureus* and MRSA to direct contact with the specimen surface. The number of bacterial colonies was quantified, and the average number of colonies for the triplicate implants at 0 hr and 24 hr was used to evaluate the antibacterial activity. Remarkably, all test samples exhibited a distinct antibacterial effect against both *S. aureus* and MRSA after 24 hours of incubation (Fig 7). 3D-printed Ti-alloy samples demonstrated an antibacterial effect, resulting in a bacterial reduction of approximately 98.8% for *S. aureus* and 90.32% for MRSA. However, TNT-A specimens yielded superior results in terms of antibacterial effect compared to 3D-printed Ti-alloy, with reductions of more than 99.99% and 99.82% for *S. aureus* and MRSA, respectively. Both groups of I-TNTs specimens (I-TNT A5 and I-TNT B5) inhibited bacterial growth at 24 hours (Table 3). The antibacterial effectiveness value for all specimens exceeded 2.4 for *S. aureus*, while I-TNT displayed an excellent antibacterial effectiveness value of more than 3.3 for MRSA (Table 4).

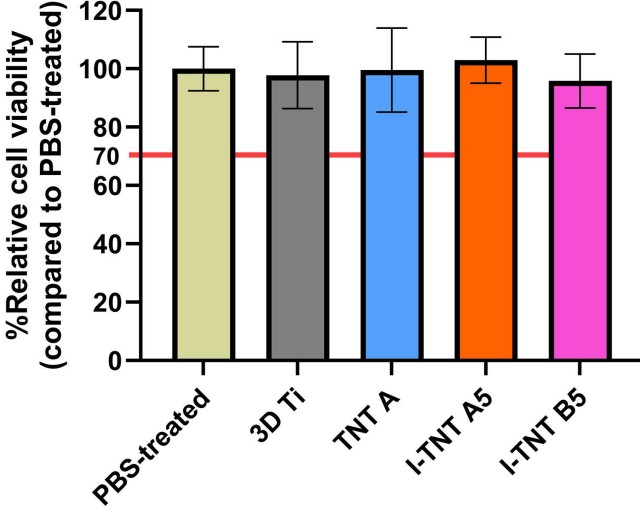

**Fig 5. Relative cell viability (%) of MC3T3-E1 cells after treated with liquid extracts from I-TNTs samples for 24 hr.** Data are expressed as mean ± SD. One-way ANOVA was used to determine statistically significant difference (n = 6).

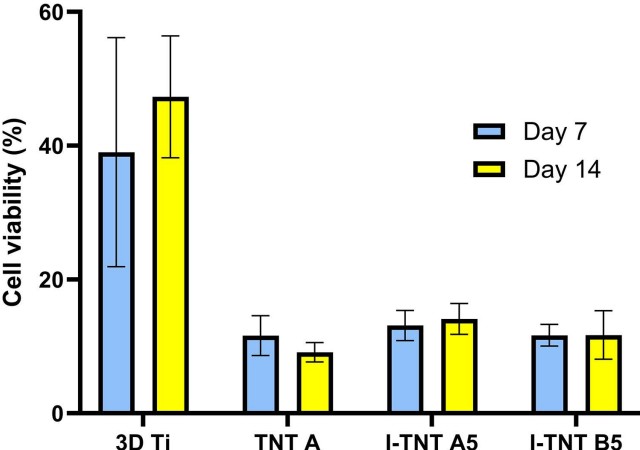

**Fig 6. Relative cell viability (%) of MC3T3-E1 cells after being exposed directly to the surface of I-TNTs for 7 days and 14 days.** Data are expressed as mean ± SD. Unpaired t-test was used to determine statistically significant difference within the same group and one-way ANOVA was used to determine statistically significant difference between each group (n = 3).

## 4. Discussion

Iodine-supported titanium implant with three distinct characteristics was successfully synthesized. Firstly, the use of TNT has significantly improved the drug-release efficiency of titanium-based implants [33]. To achieve this, we employed the ECA method to create a well-ordered array of TNT structures on the 3D printed titanium substrate [14]. These TNT structures served as drug reservoirs for the iodine (Fig 3). Secondly, we implemented electrophoretic deposition with an aqueous solution of KI to load iodine onto the TNT structures [34]. Finally, we selected 3D printed titanium as the base substrate for our implant.

The TNTs generated in this study are estimated to be approximately 6 μm in length (Table 2), which comparatively long. Delamination is more likely to occur in longer nanotubes, particularly when subjected to mechanical friction during implant [35]. It weakens the TNTs adhesion, bone integration and osteogenic signaling, thereby potentially compromising the performance of the implants, especially under cyclic loading. The remains of the TNTs may delay healing due to inflammation and the exposed implant surface becomes susceptible to bacterial colonization. The TNTs' adhesion strength to the substrate is greatly improved by heat treatment, which turns the amorphous phase of the TNTs into Anatase [36]. Optimizing the anodization parameters and pre-polishing the substrate are also reported to enhance adhesion strength [37].

The binding mode of iodine ions to titania nanotubes primarily involves chemical bonding rather than mere physical adsorption. Research indicates that iodine can form stable interactions with the titanium dioxide ($TiO_2$) surface, enhancing its photocatalytic properties and antibacterial activity. Iodine can form chemical bonds with titania, as demonstrated in iodine-doped titanium dioxide where it is incorporated into the $TiO_2$ lattice. This incorporation, involving the formation of I–O–Ti and I–O–I structures, enhances visible light absorption and photocatalytic activity [38]. In the context of carbon nanotubes, iodine can covalently bond to the surface, maintaining the electronic properties of the nanotubes. This suggests that similar covalent interactions could occur with titania nanotubes, although this is not explicitly confirmed in the provided contexts [39]. While some studies suggest that certain molecules can be physically adsorbed onto titania nanotubes [40], the evidence for iodine indicates a preference for chemical bonding, which is crucial for its enhanced functional properties. This distinction highlights the importance of chemical interactions in optimizing the performance of titania-based materials.

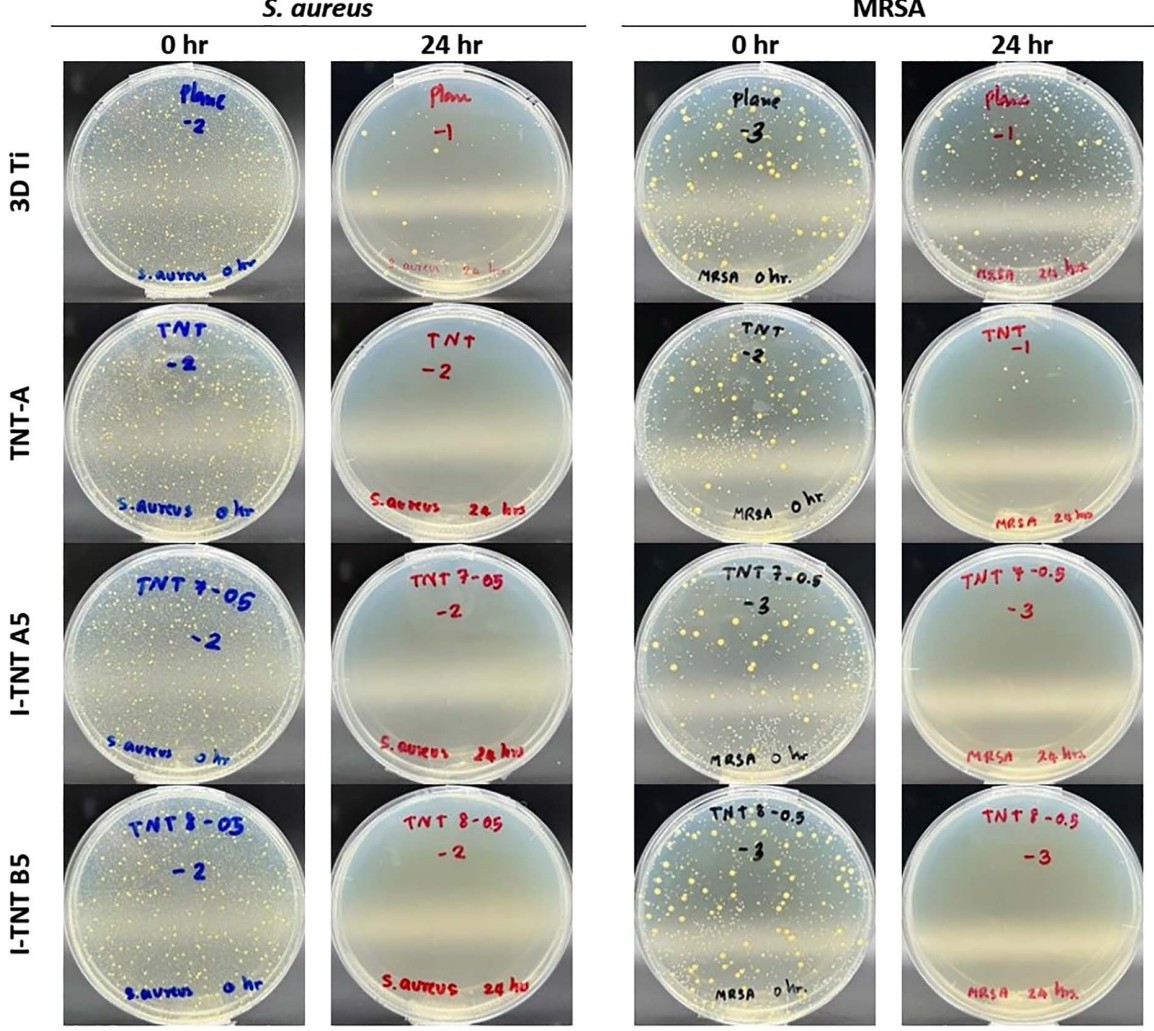

**Fig 7. Plate colony counts show *in vitro* antibacterial activity of 3D Titanium, TNT-A, I-TNT A5, and I-TNT B5 on *S. aureus* and MRSA after 24 hours of incubation.**

This study investigated the amount of iodine released from TNT on 3D-printed Ti-6Al-4V specimens. The release of iodine from TNT occurred continuously during the initial 7-day period, after which it reached a plateau phase. Following this, the release of iodine is observed to occur gradually until the 28th day, as depicted in Fig 4. It should be noted that the initial burst release characteristic, which is commonly observed in other studies involving antibiotic or antimicrobial coatings, is not observed in this particular study [41–43]. On the 7th day, the amount of iodine released reaches the minimum inhibitory concentration (MIC) level for various bacteria, including *S. aureus, P. aeruginosa, P. vulgaris, K. aerogenes,* and *E. coli* [44]. The sustained release capability of iodine, as observed in our study, is consistent with the findings of a previous study demonstrating the long-term antibacterial activity of iodine-supported titanium implants [19]. A similar report, demonstrates electrophoretic deposition of iodine into TiO$_2$-NTs, showing sustained release in saline at 37 °C, which is about a 20% drop within 2 weeks and 80% by 1 year. Under physiological conditions, this suggests that the release is controlled by diffusion [45]. We have observed three primary factors that impact the profile of iodine release. Initially,

**Table 3. The number of viable bacteria and percent reduction for *S. aureus* and MRSA.**

| Specimens | *S. aureus* | | | MRSA | | |
|---|---|---|---|---|---|---|
| | 0 hour (CFU/cm$^2$) | 24 hours (CFU/cm$^2$) | Percent reduction (%) | 0 hour (CFU/cm$^2$) | 24 hours (CFU/cm$^2$) | Percent reduction (%) |
| 3D Ti | 9.2±0.4 x10$^4$ | 1.1±0.1 x10$^3$ | 98.8 | 9.4±0.5 x10$^4$ | 9.1±0.1 x10$^3$ | 90.32 |
| TNT-A | 9.3±0.2 x10$^4$ | <4.55 | >99.99 | 9.2±0.3 x10$^4$ | 1.8±0.2 x10$^2$ | 99.82 |
| I-TNT A5 | 9.0±0.4 x10$^4$ | <4.55 | >99.99 | 9.3±0.3 x10$^4$ | <4.55 | >99.99 |
| I-TNT B5 | 9.0±0.7 x10$^4$ | <4.55 | >99.99 | 9.2±0.6 x10$^4$ | <4.55 | >99.99 |

**Table 4. Antibacterial activity value of titanium specimens to *S. aureus* and MRSA.**

| Specimens | *S. aureus* ATCC 6538 | MRSA DMST 20654 |
|---|---|---|
| TNT-A | >2.4 | 1.7 |
| I-TNT A5 | >2.4 | >3.3 |
| I-TNT B5 | >2.4 | >3.3 |

the significance of TNT fabrication is evident as the sustain-release of iodine is longer in the initial release profile when TNT is present on the surface (24 hours VS 7 days). This is in comparison to a recent study by Hu [46], which examined iodine-doped 3D-printed titanium without the fabrication of TNT on the surface. The prolonged release of iodine from TNT may be attributed to the electrostatic interactions between the oxide surface and the drug molecules [47]. Conversely, as-printed 3D-printed Ti-6Al-4V exhibits a very low cumulative release of iodine, suggesting that iodine cannot efficiently adsorb onto the surface without TNT. This is consistent with previous literature that demonstrated a low loading efficiency of gentamicin on unanodized titanium with a loading yield of 0.8% [33].

Additionally, this provides evidence of iodine diffusion into the nanotube layer and being adsorbed internally. Secondly, TNT with a higher aspect ratio (AR) (I-TNT A samples) tends to exhibit a higher cumulative release profile. This theory is supported by previous publications that state the aspect ratio is the most influential parameter in drug release among all the dimensions of TNT [33,48]. Lastly, specimens supported by iodine using a higher concentration of KI demonstrate a greater cumulative release of iodine. This is hypothesized to occur due to the increased concentration of iodine on the surface, resulting in a concentration gradient between the nanotube surface and the surrounding medium. Consequently, a considerably higher amount of iodine is released [49].

The preliminary kinetic analysis was performed using the iodine release profiles obtained from all four I-TNT formulations and the calculated total theoretical iodine content. Upon comparison of the correlation coefficients ($R^2$), the Higuchi model was identified as the best statistical fit for the experimental data (Data in S3 File). The adherence of the release data to the Higuchi model implies a diffusion-controlled sustained-release profile. Mechanistically, this demonstrates that the primary governing process is Fickian diffusion. Iodine molecules move out of the I-TNT matrix driven by a concentration gradient, passing through pores formed as the surrounding medium penetrates the TNT. This characteristic of a diffusion-controlled sustained release is particularly advantageous for controlled-release formulations, enabling more predictable dosing and subsequently reducing the risk of concentration-related side effects associated with plasma concentration peaks.

A crucial factor in assessing biocompatibility is cytotoxicity, where cultured cells are directly exposed to the test material or its extract. Our study employed the MTT assay, a common method for evaluating cell viability and proliferation. The results indicate no cytotoxicity from the iodine extract obtained from 3D-printed titanium specimens, with cell viability exceeding 70% (Fig 5). The MTT assay does not demonstrate any noteworthy distinction in cell viability between day 7 and day 14 regarding the proliferation test, as depicted in Fig 6. However, 3D-printed Ti-6Al-4V exhibits a noticeably

higher percentage of cell viability than TNT specimens. It is possible that the lower viability observed on TNT might be due to reduced cell adhesion. Previous research suggests that the size of MC3T3-E1 cells (20–50 μm) [50] may not be optimal for adhesion to the nanoscale features of TNT surfaces [51]. These findings propose that not only do our iodine-supported specimens possess cytocompatibility on their surface, but they also do not elicit cytotoxic effects when their extract is examined. This is in accordance with prior literature that indicates iodine-supported titanium implants exhibit favorable biocompatibility when tested with the V79 cell line (Chinese hamster fibroblasts) [52] and MC3T3-E1 preosteoblasts [46].

Iodine-supported specimens exhibit a remarkable antibacterial characteristic towards *S. aureus* and MRSA, as evidenced by a reduction in bacterial count exceeding 99.9% using the plate colony count method, which is consistent with previous research [46,52,53]. The antiseptic properties of iodine arise from the presence of molecular iodine ($I_2$) that is distributed on the surface of the specimens [54]. Aqueous solutions of iodine are inherently unstable and may contain various ions depending on the prevailing conditions. Out of the ten different forms of iodine found in aqueous solutions, only three possess antiseptic properties, namely $I_2$ (hydrated iodine), HOI (hypoiodous acid), and $H_2OI^+$ (hydrated iodine cation). The proposed reaction involving molecular iodine in water can be represented as $I_2 + H_2O \rightarrow HOI + H^+ + I^-$. The iodine on the surface is believed to be in the form of iodophor, a complex that significantly improves water solubility compared to $I_2$ [55]. The dominant speciation of iodine under physiological conditions is primarily characterized by the presence of iodide ($I^-$) and iodate ($IO3^-$). While $I_2$ and HOI can exist as transient or reactive intermediates, their equilibrium concentrations are generally very low due to rapid hydrolysis and subsequent reactions at neutral pH [56].

Our study also assessed the antibacterial effectiveness of as-printed 3D-printed Ti-6Al-4V samples. Diverse surface characteristics of 3D printed titanium, for example, charge density, hydrophilicity, surface roughness, surface topography, and mechanical rigidity, could potentially impact the adherence of bacteria, thereby demonstrating decreased bacterial adhesion [51]. When examining TNT specimens, their antibacterial effect is found to be higher compared to the as-printed samples, consistent with the previous research conducted by Yang et al., who compared the antibacterial effect before and after anodization, resulting in an antibacterial effect of 8.5% and 23%, respectively [57]. The nanostructure on the TNT surface may hinder bacterial attachment, potentially contributing to its antibacterial effect [58,59]. Although the cumulative release of iodine reaches the MIC level on day 7, the antibacterial property of the implant surface becomes evident within 24 hours. We can speculate that iodine-supported 3D-printed titanium can potentially prevent bacterial attachment and induce contact killing on its surface, making it an excellent property to mitigate perioperative infections typically caused by intraoperative contamination. Our findings are corroborated by a previous study that directly measured the iodine content on the implant using X-ray fluorescence spectroscopy, which reported that a minimum iodine content of approximately 3 μg/cm$^2$ completely inhibited the growth of *S. aureus* colonies after 24 hours and *E. coli* colonies after 6 hours [19]. Furthermore, the antibacterial efficacy value of I-TNTs specimens surpassed 2.0, indicating their excellent antibacterial property per the standards set by JIS Z 2801. TNT-A specimens were employed as a reference cohort for the assessment of antimicrobial effectiveness and cytotoxicity due to its high AR characteristic, thereby serving as a good representative for TNT specimens. High AR nanostructures represent not only a significant parameter concerning drug release but also possess the capability to modulate cellular adhesion through mechanotransduction pathways. This phenomenon yields a synergistic effect characterized by osteogenic and bactericidal properties, as documented in prior investigations regarding elevated AR nanopillars situated on titanium surfaces (black Ti) [60].

Based on the results from our investigation, an elevated aspect ratio (AR) in conjunction with increased potassium iodide (KI) concentration are identified as advantageous factors for the cumulative release of iodine. Nevertheless, a higher AR does not demonstrate a significant superiority over the lower AR cohort concerning cytotoxicity or antibacterial efficacy, as it yields commendable outcomes across both evaluative measures. Future research endeavors may concentrate on the synthesis of high AR TNT with varying concentrations of KI, utilizing our manuscript as a reference to ascertain the optimal parameters for the fabrication of iodine-doped TNT. There are multiple limitations in our investigation. First and foremost, the quantification of iodine concentration on the implant's surface was not carried out. This absence of data

prevents us from obtaining valuable insights into drug release kinetics, including the loading yield percentage of iodine, the percentage of iodine released, and the residual iodine content on the surface over time. In a previous study conducted by our team, the approximate theoretical wt.-% of iodine doped into titania nanotubes was determined. The iodine content obtained from EDS analysis (Fig 6) (0.05 wt.-%) was found to be in close agreement with this theoretical value (0.038 wt.-%) [27]. Secondly, the modified ECA process employed for iodine deposition relies on constant voltage and time, thereby hindering our ability to establish the correlation between these parameters and the amount of iodine deposition as well as the kinetics of drug release. Thirdly, the MIC of iodine should be interpreted with caution as this study provides a controlled environment and concentration of the solution in the analysis. Further in-vivo study should be performed to confirm the efficacy of this implant in terms of MIC determination. Finally, the antibacterial efficacy test was limited to a few bacterial strains. Although *S. aureus* and MRSA are significant contributors to orthopedics IAI [61], further research is needed to evaluate the implant's effectiveness against other common pathogens like *P. aeruginosa* and *Klebsiella*.

## 5. Conclusion

Considering the global emergence of antibiotic-resistant organisms, the use of iodine as a doping material presents itself as an ideal choice. This study is the first to comprehensively report on the release profile, cytotoxicity, and antimicrobial effectiveness of iodine-supported on 3D printed titanium, utilizing TNT as a dependable and reproducible drug delivery system. The sustained release of iodine over 28 days, without any initial burst release, combined with the absence of cytotoxic effects and excellent antimicrobial efficacy, establishes this novel technique as an intriguing option for preventing infections on 3D printed titanium implants.

## Supporting information

**S1 Fig. The calibration curve for iodine by ICP/MS.**
(PDF)

**S2 Fig. Relative cell viability (%) of Saos-2 cells after being treated with a positive control (SLS).**
(PDF)

**S3 File. Fitting analysis of kinetics models for cumulative iodine release profile.**
(PDF)

## Acknowledgments

The authors would like to thank the researchers of M3D laboratory, Faculty of Engineering,Chulalongkorn University and Analytical and Testing service Center (ATC), Department of Pharmacy, Faculty of Pharmacy, Mahidol University, Bangkok, Thailand.

## Author contributions

**Conceptualization:** Pariwat Taweekitikul, Saran Tantavisut.

**Funding acquisition:** Saran Tantavisut.

**Investigation:** Pariwat Taweekitikul, Abdul Azeez Abdu Aliyu, Nanthawan Jinakul, Amaraporn Wongrakpanich.

**Methodology:** Pariwat Taweekitikul, Abdul Azeez Abdu Aliyu, Nanthawan Jinakul, Amaraporn Wongrakpanich.

**Project administration:** Saran Tantavisut.

**Resources:** Jirapon Khamwannah, Amaraporn Wongrakpanich, Chedtha Puncreobutr, Boonrat Lohwongwatana.

**Supervision:** Amaraporn Wongrakpanich, Saran Tantavisut.

**Writing – original draft:** Pariwat Taweekitikul.

**Writing – review & editing:** Pariwat Taweekitikul, Nanthawan Jinakul, Amaraporn Wongrakpanich, Chedtha Puncreobutr, Boonrat Lohwongwatana, Saran Tantavisut.

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
