## [Decision Letter · Decision Letter 0]

19 May 2025

PLOS ONE

Dear Dr. Tantavisut,

Thank you for submitting your manuscript to PLOS ONE. After careful consideration, we feel that it has merit but does not fully meet PLOS ONE’s publication criteria as it currently stands. Therefore, we invite you to submit a revised version of the manuscript that addresses the points raised during the review process.

--It will be helpful to include EDX/ XPS or PIXE elemental analysis data for validation of concentration of Iodine. 

We look forward to receiving your revised manuscript.

Kind regards,

Tapash Ranjan Rautray

Academic Editor

PLOS ONE

Journal Requirements:

4. Thank you for stating the following financial disclosure: [Ratchadapiseksompotch Research fund, Chulalongkorn University]. 

5. We note that your Data Availability Statement is currently as follows: [All relevant data are within the manuscript and its Supporting Information files.]

Additional Editor Comments

Your manuscript has scientific potential that should be worth published in our esteemed journal PlosOne. However, I suggest you to kindly modify your paper based on the Reviewers' comments so as to align the paper in a more logical and scientific way.

Reviewers' comments:

Reviewer's Responses to Questions

**Comments to the Author**

1. Is the manuscript technically sound, and do the data support the conclusions?

Reviewer #1: Yes

Reviewer #2: Yes

2. Has the statistical analysis been performed appropriately and rigorously?

Reviewer #1: Yes

Reviewer #2: N/A

3. Have the authors made all data underlying the findings in their manuscript fully available?

Reviewer #1: Yes

Reviewer #2: Yes

4. Is the manuscript presented in an intelligible fashion and written in standard English?

Reviewer #1: Yes

Reviewer #2: No

Reviewer #1: The manuscript “PONE-D-25-15380” titled “Biocompatibility and Antimicrobial Efficacy of Iodine-supported Titania Nanotubes on 3D-Printed Ti-6Al-4V Implants” presents a novel strategy that integrates iodine into titania nanotubes (TNTs) on 3D-printed Ti-6Al-4V implants. The combination of 3D printing, electrochemical anodization (ECA)-fabricated TNTs, and iodine loading is innovative and aligns well with current clinical demands for infection-resistant implants. However, to meet the publication standards of PLOS ONE, the manuscript would benefit from clarification and enhancement in several areas, including the justification of experimental conditions, iodine quantification, and more rigorous analytical characterizations.

Q1. While the six formulations are well-documented, the rationale for selecting specific KI concentrations (0.1 and 0.5 g/L) and ECA durations (210 and 240 minutes) remain unclear. The authors are encouraged to provide justification for these parameters, either from previous literature or preliminary optimization data.

Q2. A critical limitation of the study is the absence of quantitative analysis of iodine loading (e.g., via XPS or EDS). Without precise quantification of iodine content on the TNT surface, interpretation of the release kinetics remains incomplete. This limitation is briefly acknowledged in the discussion, but it significantly weakens the conclusions regarding loading efficiency and capacity.

Q3. The MTT assay was appropriately conducted according to ISO 10993-5 standards, and the results are clearly presented. However, the inclusion of additional cytotoxicity assays such as Live/Dead staining or LDH assay would further support the biocompatibility claims, particularly in the context of long-term implant-tissue interaction.

Q4. Several statistical comparisons report p-values generically (e.g., “p < 0.001”). Providing exact p-values for all statistically analyzed data would improve transparency and reproducibility.

Q5. The FE-SEM image (page 22) successfully demonstrates the TNT structure. However, the relatively rough surface morphology of the experimental group raises concerns about stable cell attachment. It is recommended that the authors include additional FE-SEM images of the surfaces after cell culture to provide visual confirmation of cellular adhesion and morphology.

Q6. The TNTs generated in this study are estimated to be approximately 6 µm in length. Longer nanotubes are generally more susceptible to delamination, especially when subjected to mechanical friction during implantation. Given the screw-type design of the implant, the authors should address potential clinical risks of delamination and propose strategies to mitigate this mechanical vulnerability.

Q7. The discussion regarding iodine species (I₂, HOI, H₂OI⁺) is informative and appreciated. To further strengthen this section, the authors should consider referencing studies on iodine diffusion within nanotubular structures and the dominant speciation of iodine under physiological conditions.

Reviewer #2: Overall Evaluation:

This study successfully synthesized iodine-loaded titanium dioxide nanotubes (I-TNTs) on the surface of 3D-printed Ti-6Al-4V implants via electrochemical anodization and systematically evaluated their drug release properties, antibacterial efficacy, and cytocompatibility. It represents the first application of iodine-loaded TNTs on 3D-printed Ti-6Al-4V implants, filling a gap in this field and providing a novel approach to combating implant-associated infections (IAI). The research design is sound, the experimental data are comprehensive, and the conclusions are innovative with potential clinical value. However, some experimental details need further clarification, and the discussion section should strengthen the mechanistic interpretation.

Specific Comments:

1. Research Background and Significance

The introduction clearly outlines the advantages of 3D-printed titanium alloy implants and the associated infection risks, logically proposing the innovative strategy of combining iodine with TNTs. However, some cited references (e.g., Refs. 23 and 24, published in 1975 and 1987, respectively) are outdated. It is recommended to update these with more recent studies (within the last five years) to enhance the timeliness of the discussion.

2. Experimental Design

The study comprehensively evaluates drug release, cytotoxicity, and antibacterial activity, aligning with the standards for biomaterials research.

Suggestions for Improvement:

Iodine Loading Mechanism: The "Modified ECA process" using KI is mentioned, but the binding mode of iodine ions to TNTs (e.g., physical adsorption or chemical bonding) is not explicitly described.

In Vitro-In Vivo Correlation: The in vitro release curve shows that the MIC is reached at 7 days, but the antibacterial tests only assess 24-hour efficacy. It is recommended to supplement time-kill curves (e.g., at 1, 3, and 7 days) to clarify long-term antibacterial performance.

3. Results and Analysis

Drug Release Curve (Figure 4):

The sustained-release characteristics are clearly described, but the release kinetics model (e.g., zero-order, first-order, or Higuchi model) is not analyzed. Fitting analysis should be added to elucidate the release mechanism.

4. Formatting and Language

Ensure consistent formatting of references.

Strengthen the discussion section by enhancing mechanistic explanations and linking findings to clinical significance.

**Do you want your identity to be public for this peer review?** For information about this choice, including consent withdrawal, please see our Privacy Policy

Reviewer #1: No

Reviewer #2: No

---

## [Author Response · Author response to Decision Letter 1]

3 Jul 2025

Response to Editor's Comments

Q. No. 1 It will be helpful to include EDX/ XPS or PIXE elemental analysis data for validation of concentration of iodine.

Response: We appreciate the Editor's comment and the opportunity given to improve the quality of this paper. However, the authors would like to clarify that full characterization of iodine-doped TNTs, including the EDX and XPS elemental analysis, is presented in our recently published paper [1] (XPS: Figure 4(b) and Figure 5, Page 7-8, and EDX: Figure 6, Page 9). This statement is added in section 3.2 of the current paper. The published paper is also cited in this section. (Location: 3.2 Iodine release profile from the I-TNTs specimens)

Reference

[1] Taweekitikul P, Aliyu A, Decha‐Umphai D, Tantavisut S, Khamwannah J, Puncreobutr C, et al. Synthesis, characterization, and interfacial adhesion of titania iodine‐doped nanotubes architectures on additively manufactured Ti‐6Al‐4V implant. Materialwissenschaft und Werkstofftechnik. 2025;56(3):438-54.

Response to Reviewer's Comments

We appreciate the reviewers' careful reading of our manuscript and their thoughtful comments and critiques. The reviewers made several suggestions to improve the manuscript. Enclosed please find our step-by-step responses to address the points raised by the reviewer.

Reviewer 1

Q. No. 1 While the six formulations are well-documented, the rationale for selecting specific (a) KI concentrations (0.1 and 0.5 g/L) and (b) ECA durations (210 and 240 minutes) remains unclear. The authors are encouraged to provide justification for these parameters, either from previous literature or preliminary optimization data.

Response: (a) The rationale for using specific KI concentrations (0.1 and 0.5 g/L)

The minimum bactericidal concentration (MBC) of povidone iodine against specific strains of Staphylococcus aureus and Staphylococcus epidermidis is a crucial parameter in evaluating its antiseptic efficacy. The previous publication reported that the minimum bactericidal concentrations (MBC) of povidone iodine against S. aureus and S. epidermidis are 1.32 g/L, and complete cytotoxicity to human cells occurs at 2 g/L [1].

Based on general knowledge, povidone-iodine solutions typically contain 1% available iodine, while KI contains 2% iodine. To achieve the same iodine content as 1.32 g/L of povidone-iodine, approximately 0.66 g/L of KI is required, ensuring that KI does not exceed 1 g/L to avoid potential cytotoxicity to human cells. In this study, we selected KI concentrations of 0.1 g/L and 0.5 g/L for preliminary analysis. We planned to increase the KI concentration up to 1 g/L if the antibacterial efficacy results were unsatisfactory.

(b) The rationale for using specific ECA durations (210 and 240 minutes)

Prior publications indicate that AR is the most influential parameter for drug release among all the dimensions of TNT. According to Miyabe et al., different lengths and diameters of nanotubes could be manipulated to control drug loading and release rates. The higher the aspect ratio, the more efficient the drug delivery system can be, as the larger surface area permits enhanced drug loading capabilities [2]. A previous study conducted by our team [3] demonstrated that ECA durations of 210 and 240 minutes resulted in the highest aspect ratio (AR) for titania nanotubes (Figure 2, Page 6 in the previous manuscript). Thus, these specific ECA durations were selected for the current investigation.

These statements were summarized and incorporated into the modified manuscript. (Location: 2.1 Preparation of iodine-supported TNT)

Q. No. 2 A critical limitation of the study is the absence of quantitative analysis of iodine loading (e.g., via XPS or EDS). Without precise quantification of iodine content on the TNT surface, interpretation of the release kinetics remains incomplete. This limitation is briefly acknowledged in the discussion, but it significantly weakens the conclusions regarding loading efficiency and capacity.

Response: Thank you for this comment. The authors would like to clarify that the quantitative analysis of iodine content in the TNTs is out of the scope of this paper. However, in our recently published article [2], the approximate theoretical wt. % of iodine doped into the titania nanotubes was determined. The iodine content obtained from EDS analysis (Figure 6, Page 9) (0.05 wt. %) is found to be in agreement with the theoretical value (0.038 wt. %) (Page 5).

Q. No. 3 The MTT assay was appropriately conducted according to ISO 10993-5 standards, and the results are clearly presented. However, the inclusion of additional cytotoxicity assays, such as Live/Dead staining or LDH assay, would further support the biocompatibility claims, particularly in the context of long-term implant-tissue interaction.

Response: Additional in vitro cytotoxicity assays (e.g., Live/Dead staining or LDH assay) and in vivo biocompatibility tests are planned for future investigations to support long-term implant-tissue interaction. However, the present study specifically aims to establish the perioperative cytotoxicity of this implant. Consequently, we report only the short-term relative cell viability of MC3T3-E1 cells following treatment with liquid extracts from I-TNTs samples for 24 hours and after direct exposure to the surface of I-TNTs for up to 14 days.

Q. No. 4 Several statistical comparisons report p-values generically (e.g., "p < 0.001"). Providing exact p-values for all statistically analyzed data would improve transparency and reproducibility.

Response: Replace p < 0.001 with the exact p-values

1. The cumulative iodine release profiles were significantly higher in the I-TNTs groups compared to the control group (p = 0.000).

2. The first significant difference between I-TNT A1 and I-TNT A5 was observed at 24 hours (p = 0.014), while I-TNT B1 and I-TNT B5 showed significant differences at the final time-point of 672 hours (p = 0.000).

The exact values of the p-values were written in the manuscript as suggested.

Q. No. 5 The FE-SEM image (page 22) successfully demonstrates the TNT structure. However, the relatively rough surface morphology of the experimental group raises concerns about stable cell attachment. It is recommended that the authors include additional FE-SEM images of the surfaces after cell culture to provide visual confirmation of cellular adhesion and morphology.

Response: While cell attachment is indeed an essential characteristic for orthopedic implants, it is important to note that for this specific I-TNT structure, our primary focus is on its antimicrobial properties, achieved through the controlled release of iodine. The detailed cell adhesion morphology, as observed by FE-SEM, will be presented in a subsequent study.

Q. No. 6 The TNTs generated in this study are estimated to be approximately 6 µm in length. Longer nanotubes are generally more susceptible to delamination, especially when subjected to mechanical friction during implantation. Given the screw-type design of the implant, the authors should address potential clinical risks of delamination and propose strategies to mitigate this mechanical vulnerability.

Response: The potential clinical risks of delamination and proposed strategies to mitigate the effects of very long (≥ 6 µm) TNTs and implants were clearly stated and supported by the literature in the discussion section. (Location: 2nd paragraph of Discussion)

Q. No. 7 The discussion regarding iodine species (I₂, HOI, H₂OI⁺) is informative and appreciated. To further strengthen this section, the authors should consider referencing studies on iodine diffusion within nanotubular structures and the dominant speciation of iodine under physiological conditions.

Response: More information related to iodine diffusion within TNT structures was added in the discussion section (Location: 4th paragraph of Discussion). The dominant speciation of iodine under physiological conditions is primarily characterized by the presence of iodide (I−) and iodate (IO3−). While I₂ and HOI can exist as transient or reactive intermediates, their equilibrium concentrations are generally very low due to rapid hydrolysis and subsequent reactions at neutral pH [4] This information was also cited as far as suggested. (Location: 4th paragraph of Discussion)

References

1. Van Meurs S, Gawlitta D, Heemstra K, Poolman R, Vogely H, Kruyt M. Selection of an optimal antiseptic solution for intraoperative irrigation: an in vitro study. JBJS. 2014;96(4):285-91.

2. Miyabe S, Fujinaga Y, Tsuchiya H, Fujimoto S. TiO2 nanotubes with customized diameters for local drug delivery systems. Journal of Biomedical Materials Research Part B: Applied Biomaterials. 2024;112(7):e35445.

3. Taweekitikul P, Aliyu A, Decha‐Umphai D, Tantavisut S, Khamwannah J, Puncreobutr C, et al. Synthesis, characterization, and interfacial adhesion of titania iodine‐doped nanotubes architectures on additively manufactured Ti‐6Al‐4V implant. Materialwissenschaft und Werkstofftechnik. 2025;56(3):438-54.

4. Espino-Vázquez AN, Rojas-Castro FC, Fajardo-Yamamoto LM. Implications and practical applications of the chemical speciation of iodine in the biological context. Future Pharmacology. 2022;2(4):377-414.

Reviewer 2

Q. No. 1 Research Background and Significance

The introduction clearly outlines the advantages of 3D-printed titanium alloy implants and the associated infection risks, logically proposing the innovative strategy of combining iodine with TNTs. However, some cited references (e.g., Refs. 23 and 24, published in 1975 and 1987, respectively) are outdated. It is recommended to update these with more recent studies (within the last five years) to enhance the timeliness of the discussion.

Response: Refs 23 and 24 were replaced with the most recent references (refs 24 and 25, respectively) as suggested.

Q. No. 2 The study comprehensively evaluates drug release, cytotoxicity, and antibacterial activity, aligning with the standards for biomaterials research.

Suggestions for Improvement:

(a) Iodine Loading Mechanism: The "Modified ECA process" using KI is mentioned, but the binding mode of iodine ions to TNTs (e.g., physical adsorption or chemical bonding) is not explicitly described.

(b) In Vitro-In Vivo Correlation: The in vitro release curve shows that the MIC is reached at 7 days, but the antibacterial tests only assess 24-hour efficacy. It is recommended to supplement time-kill curves (e.g., at 1, 3, and 7 days) to clarify long-term antibacterial performance.

Response: (a) Binding mode of iodine ions to TNTs

The binding mode of iodine ions to titania nanotubes primarily involves chemical bonding rather than mere physical adsorption. Research indicates that iodine can form stable interactions with the titanium dioxide (TiO2) surface, enhancing its photocatalytic properties and antibacterial activity.

Iodine can form chemical bonds with titania, as demonstrated in iodine-doped titanium dioxide, where it is incorporated into the TiO2 lattice. This incorporation, involving the formation of I–O–Ti and I–O–I structures, enhances visible light absorption and photocatalytic activity [1].

In the context of carbon nanotubes, iodine can covalently bond to the surface, maintaining the electronic properties of the nanotubes. This suggests that similar covalent interactions could occur with titania nanotubes, although this is not explicitly confirmed in the provided contexts [2].

Considering the physical adsorption, it typically involves weak van der Waals forces and does not result in significant changes to the electronic structure or the crystallography of the substrate. In contrast, the literature describes scenarios where iodine not only adsorbs onto TiO2 nanotubes but also integrates into the lattice, modifying electronic properties and stability, which are characteristics typical of chemical bonding [3].

While some studies suggest that certain molecules can be physiosorbed onto titania nanotubes [4], the evidence for iodine indicates a preference for chemical bonding, which is crucial for its enhanced functional properties. This distinction underscores the significance of chemical interactions in enhancing the performance of titania-based materials. (Location: 3rd paragraph of Discussion)

Time-kill curve

This particular implant primarily aims to prevent implant-associated infection resulting from perioperative contamination. Consequently, it adheres to the standards set by JIS Z 2801, where the antibacterial activity value is calculated based on 24-hour efficacy. Data from a time-kill curve extending up to 7 days would provide valuable insights, which we intend to present in a future study.

Q. No. 3 Drug Release Curve (Figure 4):

The sustained-release characteristics are clearly described, but the release kinetics model (e.g., zero-order, first-order, or Higuchi model) is not analyzed. Fitting analysis should be added to elucidate the release mechanism.

Response: The Higuchi model most accurately describes the observed release profile. This profile exhibited an initial burst release, likely due to surface-adsorbed iodine, succeeded by a slower, diffusion-controlled release from the I-TNT matrix. The cumulative release gradually increased following a square root of time behavior, as opposed to a linear progression.

Q. No. 4 Formatting and Language

Ensure consistent formatting of references.

Strengthen the discussion section by enhancing mechanistic explanations and linking findings to clinical significance.

Link findings to clinical significance.

Response: The manuscript is formatted according to the journal's guidelines. The manuscript was further proofread. The discussion section was strengthened by adding the potential clinical risks of TNT delamination and proposing strategies to mitigate its effects, especially when TNT is too long (≥ 6 µm).

References

1. He Z, Yu Y, Wang D, Tang J, Chen J, Song S. Photocatalytic reduction of carbon dioxide using iodine-doped titanium dioxide with high exposed {001} facets under visible light. RSC Advances. 2016;6(28):23134-40.

2. Coleman KS, Chakraborty AK, Bailey SR, Sloan J, Alexander M. Iodination of single-walled carbon nanotubes. Chemistry of materials. 2007;19(5):1076-81.

3. Yang X, Chen N-F, Huang X-L, Lin S, Chen Q-Q, Wang W-M, et al. Iodine-doped TiO2 nanotube coatings: a technique for enhancing the antimicrobial properties of titanium surfaces against Staphylococcus aureus. Journal of Orthopaedic Surgery and Research. 2023;18(1):854.

4. Chen Q, Jia Y, Liu S, Mogilevsky G, Kleinhammes A, Wu Y. Molecules immobilization in titania nanotubes: a solid-state NMR and computational chemistry study. The Journal of Physical Chemistry C. 2008 Nov 6;112(44):17331-5.

---

## [Decision Letter · Decision Letter 1]

24 Oct 2025

PONE-D-25-15380R1

Dear Dr. Tantavisut,

Thank you for submitting your manuscript to PLOS ONE. After careful consideration, we feel that it has merit but does not fully meet PLOS ONE’s publication criteria as it currently stands. Therefore, we invite you to submit a revised version of the manuscript that addresses the points raised during the review process.

We look forward to receiving your revised manuscript.

Kind regards,

Tapash Ranjan Rautray 

Academic Editor

PLOS ONE

Journal Requirements:

Additional Editor Comments:

Since the Reviewer 3 has given comments with Minor revision decision, I want the authors to have a minor revision of the manuscript as per the comments of the Reviewer # 3. The comments as as follows.

Reviewers' comments:

Reviewer's Responses to Questions

**Comments to the Author**

Reviewer #3: (No Response)

2. Is the manuscript technically sound, and do the data support the conclusions?

Reviewer #3: Yes

3. Has the statistical analysis been performed appropriately and rigorously?

Reviewer #3: Yes

4. Have the authors made all data underlying the findings in their manuscript fully available?

Reviewer #3: Yes

5. Is the manuscript presented in an intelligible fashion and written in standard English?

Reviewer #3: Yes

Reviewer #3: The authors fabricated titania nanotubes (TNTs) loaded with iodine as a potential orthopedic implant material. The materials were characterized for iodine release, cell cytotoxicity, and antimicrobial properties. The study is clear, well described, and the conclusions are supported by data and analysis. Minor comments are below.

Minor comments:

The authors characterize the iodine release profile from I-TNT samples in Section 3.2. Statistical tests on the cumulative iodine release at different time points were performed. A more rigorous approach to characterize release profiles would be to fit kinetic models and compare parameters, rather than the present approach. A prior reviewer suggested that the authors perform kinetic modeling, and the authors have responded that “The Higuchi model most accurately describes the observed release profile…”, but no fitting analysis is found in the manuscript or supporting information. Some kinetic modeling may enhance the manuscript’s discussion of the iodine release mechanism via diffusion through TNT and may offer predictions for long term, in vivo antimicrobial activity.

The antimicrobial data is presented in Tables 3 and 4. Certain conditions are quite effective and results in measurements of “<4.55” which this reviewer assumes to the be limit of detection for this study and specific assay. For transparency, the authors should document the dilution conditions and any other relevant data used to determine this limit of detection.

For Fig. 5, 6, 7 and Tables 3 and 4, the sample naming convention seems to have changed. Is “TNT-A” the sample described as “I-TNT A1” in Table 1?

**Do you want your identity to be public for this peer review?** For information about this choice, including consent withdrawal, please see our Privacy Policy

Reviewer #3: No

---

## [Author Response · Author response to Decision Letter 2]

5 Dec 2025

Response to Reviewer #3’s Comments

1. The authors characterize the iodine release profile from I-TNT samples in Section 3.2. Statistical tests on the cumulative iodine release at different time points were performed. A more rigorous approach to characterize release profiles would be to fit kinetic models and compare parameters, rather than the present approach. A prior reviewer suggested that the authors perform kinetic modeling, and the authors have responded that “The Higuchi model most accurately describes the observed release profile…”, but no fitting analysis is found in the manuscript or supporting information. Some kinetic modeling may enhance the manuscript’s discussion of the iodine release mechanism via diffusion through TNT and may offer predictions for long term, in vivo antimicrobial activity.

Response: The preliminary fitting analysis was performed using the following sequential steps:

1. As previously stated, a limitation of this study was the quantification of the total iodine content directly on the implant surface. Consequently, we relied on the theoretical iodine loading within the titania nanotubes to define 100% release. To achieve this, the total theoretical iodine content of the I-TNT samples was first calculated. Based on the established formulation [1], the theoretical iodine content was 0.038 wt.-%. Using the average weight of the I-TNT samples (768.53 ± 20.71 mg (detailed in the minimal data set), the total theoretical iodine content was calculated to be

0.038x768.53x1000/100=292.04 µg

2. The percentage of cumulative iodine release at each sampling point was subsequently calculated by normalizing the measured iodine content to the total theoretical iodine content.

3. To identify the release kinetics, the experimental data were fitted to several key kinetic models, including the Zero-order, First-order, Higuchi, Hixson-Crowell, and Korsmeyer–Peppas models. The best-fit model was determined by comparing the correlation coefficient (R2) values. This fitting analysis was performed using Microsoft Excel.

The kinetic analysis was performed using the iodine release profiles obtained from all four I-TNT formulations. Upon comparison of the correlation coefficients (R2), the Higuchi model was identified as the best statistical fit for the experimental data (detailed results are provided in Supplement 3). This result strongly suggests that the drug release mechanism is consistent with the principles of the Higuchi model.

The adherence of the release data to the Higuchi model implies a diffusion-controlled sustained-release profile. This model is fundamentally based on the principle that the drug release rate is proportional to the square root of time (√t). Mechanistically, this demonstrates that the primary governing process is Fickian diffusion. Iodine molecules move out of the I-TNT matrix driven by a concentration gradient, passing through pores formed as the surrounding medium penetrates the TNT. This characteristic of a diffusion-controlled sustained release is particularly advantageous for controlled-release formulations, enabling more predictable dosing and subsequently reducing the risk of concentration-related side effects associated with plasma concentration peaks.

Reference:

[1] Taweekitikul P, Aliyu A, Decha‐Umphai D, Tantavisut S, Khamwannah J, Puncreobutr C, et al. Synthesis, characterization, and interfacial adhesion of titania iodine‐doped nanotubes architectures on additively manufactured Ti‐6Al‐4V implant. Materwiss Werksttech. 2025;56(3):438-54.

2.The antimicrobial data is presented in Tables 3 and 4. Certain conditions are quite effective and results in measurements of “<4.55” which this reviewer assumes to the be limit of detection for this study and specific assay. For transparency, the authors should document the dilution conditions and any other relevant data used to determine this limit of detection.

Response: The quantification of viable bacteria (CFU/cm2) (equation 1)

N=CxDxV/A(1)

where N: number of viable bacteria (per 1 cm2 of test piece)

C: count of colonies (average count of colonies of two petri dishes adopted)

D: dilution factor (that of dilution dispensed into petri dishes adopted)

V: volume of SCDLP broth used for washout (mL)

A: surface area of test piece that was exposed to bacteria (cm2)

C = 1 (when the count of colonies C is “<1”, C is taken as “1”), D = 1, V = 10 ml, A = 2.2 cm2.

N shall be expressed as “<4.55”.

3.For Fig. 5, 6, 7 and Tables 3 and 4, the sample naming convention seems to have changed. Is “TNT-A” the sample described as “I-TNT A1” in Table 1?

Response: The “TNT-A” in Fig. 5, 6, 7 and Tables 3 and 4 is labelled correctly.

For clarity across the results, the sample designation 'TNT-A' refers specifically to the TNT synthesized using the Electrochemical Anodization (ECA) technique. As detailed in Section 2.1 ('Preparation of iodine-supported TNT') and illustrated in Figure 1, the synthesis parameters for TNT-A involved a constant applied voltage of 60 V and a duration of 210 minutes. This specific set of parameters was selected because TNT-A represents the TNT sample exhibiting the highest aspect ratio identified in our previous publication [1]

The data presented in Figures 5, 6, and 7 and Tables 3 and 4 facilitate the comparison of TNT-A (TNT samples without iodine support) against the functionalized samples, I-TNT A5 and I-TNT B5 (iodine-supported TNT samples), specifically focusing on their respective cytotoxicity and antibacterial efficacy.

Reference:

[1] Taweekitikul P, Aliyu A, Decha‐Umphai D, Tantavisut S, Khamwannah J, Puncreobutr C, et al. Synthesis, characterization, and interfacial adhesion of titania iodine‐doped nanotubes architectures on additively manufactured Ti‐6Al‐4V implant. Materwiss Werksttech. 2025;56(3):438-54.

---

## [Editor Report · Decision Letter 2]

10 Dec 2025

Biocompatibility and Antimicrobial Efficacy of Iodine-supported Titania Nanotubes on 3D-Printed Ti-6Al-4V Implants

PONE-D-25-15380R2

Dear Dr. Tantavisut,

We’re pleased to inform you that your manuscript has been judged scientifically suitable for publication and will be formally accepted for publication once it meets all outstanding technical requirements.

Kind regards,

Tapash Ranjan Rautray

Academic Editor

PLOS One

Additional Editor Comments (optional):

Based on the comments of the Reviewers, the authors have modified their manuscript and clarified their concerns. So, I accept your manuscript in the present form.
---

## [Editor Report · Acceptance letter]

PONE-D-25-15380R2

PLOS One

Dear Dr. Tantavisut,

I'm pleased to inform you that your manuscript has been deemed suitable for publication in PLOS One. Congratulations! Your manuscript is now being handed over to our production team.

Kind regards,

on behalf of

Dr. Tapash Ranjan Rautray

Academic Editor

PLOS One